# AAV2retro Enters Axons of Passage and Extensively Transduces Corticospinal Neurons After Injection into Spinal White Matter

**DOI:** 10.3390/brainsci15101058

**Published:** 2025-09-28

**Authors:** Kazuki T. Nakashima, Shanshan Wang, Michael J. Castle

**Affiliations:** 1Department of Neurosciences, University of California-San Diego, La Jolla, CA 92093, USA; 2Department of Anesthesiology, University of California-San Diego, La Jolla, CA 92093, USA

**Keywords:** spinal cord injury, gene therapy, corticospinal tract, AAV, AAV2retro, AAVrg

## Abstract

Background: Adult neurons in the central nervous system often fail to regenerate after spinal cord injury (SCI). Regenerative gene therapies could potentially promote corticospinal axon regeneration, restoration of motor circuitry, and functional improvement after SCI, but translational methods for targeted gene delivery to corticospinal neurons are needed. AAV2retro is an engineered variant of the adeno-associated virus 2 (AAV2) capsid that demonstrates greatly enhanced retrograde transduction of projection neurons. When injected into spinal gray matter, AAV2retro retrogradely transduces neurons in the sensorimotor cortex that project to the injected spinal level. Methods: We initially hypothesized that injection of AAV2retro into the dorsal column white matter immediately rostral of a mouse cervical spinal injury would target transected axons and broadly transduce both forelimb and hindlimb corticospinal neurons. We tested this hypothesis by comparing four groups of mice treated with AAV2retro carrying the tdTomato reporter gene by (1) injection into intact C4 gray matter, (2) injection into intact C4 dorsal column white matter, (3) injection into C4 gray matter bordering a C5 dorsal column lesion, and (4) injection into C4 dorsal column white matter bordering a C5 dorsal column lesion. Results: After injection of AAV2retro into intact C4 dorsal column white matter, we observed extensive transduction of corticospinal neurons throughout both the forelimb and hindlimb sensorimotor cortical regions, and large numbers of transduced hindlimb corticospinal axons in the lumbar spinal cord. Dorsal column injections did not detectably damage the white matter beyond a narrow injection track. In contrast, after injection of intact C4 gray matter, we observed minimal labeling of neurons in the hindlimb sensorimotor cortex or corticospinal axons in the lumbar spinal cord. Conclusions: We conclude that AAV2retro can enter axons of passage in the dorsal column white matter of the spinal cord, and that injecting the cervical dorsal columns can efficiently target both forelimb and hindlimb corticospinal neurons in mice. This new approach for targeted gene delivery to corticospinal neurons could improve the safety and specificity of regenerative gene therapies for spinal cord injury.

## 1. Introduction

The corticospinal tract is a descending white matter pathway by which upper motor neurons in the brain connect to lower motor neurons and interneurons in the spinal cord [1,2]. The corticospinal tract is essential for voluntary movement of the trunk and limbs, and transection of corticospinal axons by spinal cord injury (SCI) results in a loss of motor function below the injury. Adult corticospinal axons do not typically regenerate after SCI due to the inhibitory lesion environment and failure to re-activate signaling pathways that control axon growth during development but are silenced in the adult central nervous system (CNS) [3,4,5]. Genetic therapies can re-activate cell growth signaling and promote the regeneration of adult corticospinal axons after SCI in mice, including overexpression of Akt3, Sox11, or Klf6 [6,7,8], deletion of the *PTEN* gene [9], and co-deletion of *PTEN* and either *SOCS3* or *RTN4* (Nogo) [10,11]. Such regenerative gene therapies could potentially restore motor circuitry and improve motor function after human SCI, but safer and more effective methods are needed for targeted gene delivery to corticospinal neurons.

Adeno-associated virus (AAV) vectors are commonly used for recombinant gene delivery to CNS neurons in animal models and humans [12,13]. Direct injection of AAV into the human brain can be safely performed [14,15], but this approach would require a prohibitively large number of invasive injections to treat the entire human primary motor cortex. Direct AAV injection would also result in mistargeting of additional cortical cells other than the layer 5 projection neurons that are injured by SCI, and would fail to engage corticospinal neurons outside of the primary motor cortex. An emerging alternative is intravenous infusion of new AAV capsids engineered to cross the adult human blood–brain barrier [16,17]. Although promising, this approach requires much higher doses of AAV and may be blocked by pre-existing antibodies in the blood. It is also unknown how SCI (and the resulting disruption of the blood–brain barrier) impacts the efficacy of these capsids for gene delivery to the brain. While infusion of AAV into the blood or cerebrospinal fluid could potentially treat all populations of upper motor neurons throughout the cortex, these approaches broadly target cells throughout the brain, spinal cord, and peripheral ganglia [16,17,18,19], and extensive off-target gene delivery may cause adverse effects. This could be addressed by developing small and specific promoters that restrict transcription to upper motor neurons, but such promoters identified to date are not highly specific, are untested in humans, and may not be sufficiently strong for effective human gene therapy [20,21].

Non-viral methods for nucleic acid delivery are appealing because they transiently alter gene expression and could potentially promote regeneration of injured neurons within a limited therapeutic window, but these approaches also have drawbacks that limit their use in SCI. Antisense oligonucleotides can transiently suppress one or more genes after infusion into the cerebrospinal fluid, but their efficacy may be limited by incomplete gene knockdown [22]. Lipid nanoparticles carrying mRNA can drive transient gene expression, but delivery of lipid nanoparticles across the blood–brain barrier remains challenging [23]. Similar to viral gene delivery, both antisense oligonucleotides and lipid nanoparticles broadly treat the nervous system after infusion into the cerebrospinal fluid or blood. Although the primary motor cortex could be treated by direct injection, this would require a large number of invasive brain injections. Thus, despite recent advances in viral and non-viral gene therapies, there is currently no translational method for gene delivery to corticospinal neurons that is both broad and specific. Improved methods are needed that effectively target upper motor neurons throughout the cortex and limit off-target gene delivery to other regions.

It was previously reported that injection of natural AAV serotypes 1, 2, 5, 8, or 9 into both spinal gray and white matter immediately rostral of a thoracic complete transection injury resulted in extensive retrograde transduction of neurons in the rat brainstem [24]. Because natural AAVs share conserved mechanisms for retrograde axonal transport [25,26], after injection of spinal white matter, these natural AAVs presumably entered transected axons of spinal-projecting brainstem neurons and were then transported retrogradely to the brainstem. However, retrograde transduction of upper motor neurons in the cortex was not observed, possibly because the spinal AAV injections targeted lateral and ventral white matter, but not the dorsal columns where the corticospinal tract is located.

AAV2retro is a variant of the adeno-associated virus 2 (AAV2) capsid modified by insertion of the 10-mer peptide LADQDYTKTA between N587 and R588 and by the point mutations N382D and V708I. Relative to natural AAV capsids, AAV2retro demonstrates greatly enhanced retrograde transduction of most projection neurons [27,28,29]. When injected into spinal gray matter, AAV2retro retrogradely transduces corticospinal neurons that project to the injected spinal level [28,30]. We hypothesized that directly injecting AAV2retro into the dorsal column white matter immediately rostral of a mouse cervical spinal injury would target transected corticospinal axons and broadly transduce upper motor neurons throughout the cortex. We also examined direct injection of AAV2retro into intact dorsal column white matter in uninjured mice, because to our knowledge no previous studies have tested whether AAV2retro can transduce axons of passage in spinal white matter. If effective, injection of AAV2retro into the dorsal column white matter could potentially provide a translational method for targeted gene delivery to corticospinal neurons and support the preclinical development of regenerative gene therapies for human SCI.

## 2. Materials and Methods

### 2.1. AAV Vector

AAV2retro-CAG-tdTomato-WPRE was obtained from Addgene (viral prep #59462-AAVrg). AAV was produced by triple transfection in adherent HEK 293 cells followed by iodixanol gradient ultracentrifugation. The viral genome carried a truncated 887 bp variant of the chicken beta-actin gene (CAG) promoter, a codon-diversified tdTomato protein coding sequence, the 589 bp woodchuck hepatitis virus post-transcriptional regulatory element (WPRE), and the 122 bp SV40 late polyadenylation signal. Immediately prior to injection, AAV2retro was diluted to 1 × 10^12^ vector genomes (vg)/mL in sterile 0.01 M phosphate-buffered saline (PBS) containing 0.005% poloxamer 188.

### 2.2. Mouse Surgery and Sacrifice

All animal experiments were approved by the Institutional Animal Care and Use Committee of the Department of Veterans Affairs San Diego Healthcare System and conducted in compliance with the National Institutes of Health laboratory animal care and safety guidelines. A total of N = 16 female C57/BL6J mice (4 months old) were treated in this study (N = 4 per group). Each side of the brain or spinal cord was analyzed as an anatomically independent datapoint, providing an initial sample size of N = 8 per group (see Section 3.1). A minimum sample size of N = 4 was calculated by power analysis with an alpha criterion of 0.05, 95% power, and an estimated effect size of *d* = 3 (based on the approximate number of corticospinal neurons per brain section predicted to be transduced by cervical gray matter injection, and the approximate number of additional corticospinal neurons per brain section that could potentially be transduced by targeting axons in the dorsal column white matter). It was predetermined that all mistargeted injections would be excluded: each injection track was examined by immunofluorescent and Nissl staining of the cervical spinal cord (see Section 3.4), and injections that were not accurately targeted to the gray matter or dorsal column white matter were excluded from statistical analysis. The initial sample size of N = 8 accounted for the potential exclusion of up to N = 4 injections per group while retaining the minimum sample size of N = 4 as determined by the power calculation above.

Animals were randomly assigned to each group at the time of surgery. In all mice, a laminectomy of the C4 vertebrae was performed to expose the C4 and C5 spinal cord [31]. Half the mice (N = 8) received a C5 dorsal column lesion, performed by puncturing the dura with a 27-gauge needle, then inserting a retracted tungsten wire knife (David Kopf instruments) through the puncture hole 0.3 mm lateral of the spinal midline [5,32]. The wire knife was lowered 0.5 mm beneath the spinal surface, then extended 0.6 mm in a ventromedial arc across the midline and beneath the dorsal columns. The wire knife was then gradually raised to the spinal surface while a 25-gauge blunt needle was pressed down onto the dorsal spinal cord to ensure complete bilateral transection of the dorsal columns. The wire knife was then retracted and withdrawn from the mouse, leaving the dura intact except for a small puncture hole. The other N = 8 mice did not receive a spinal cord injury.

AAV2retro injections were performed using a pulled glass micropipette with an outer diameter of 35–45 µm [33]. The micropipette was mounted on a stereotactic frame and connected with vacuum tubing to a Picospritzer II pressure injector (General Valve). All injections were performed by repeated pressure pulses at a pressure of 20 PSI and a pulse duration of 5–8 ms. Because the volume injected per pulse is potentially variable due to partial blockage of the pipette tip, the volume injected at each site was determined by visually monitoring the decrease in volume of AAV solution within the glass pipette. Immediately after SCI, half of the mice that received a C5 dorsal column lesion (N = 4) were treated by bilateral injection of AAV2retro targeting the spinal gray matter ~0.2 mm rostral of the lesion site. The dura was punctured with a 27-gauge needle and a micropipette was inserted through the puncture hole 0.5 mm lateral of the spinal midline. The micropipette was lowered to ~0.7 mm beneath the spinal surface and 200 nL of AAV2retro was injected over 1 min. The micropipette was left in place for 30 s, raised to ~0.4 mm beneath the spinal surface, and another 200 nL of AAV2retro was injected over 1 min. After 30 additional seconds, the micropipette was slowly withdrawn from the mouse and identical injections were performed on the other side of the spinal cord.

The other N = 4 mice that received a dorsal column lesion were treated by bilateral injection of AAV2retro targeting the dorsal column white matter ~0.2 mm rostral of the lesion site. The dura was punctured with a 27-gauge needle and a micropipette was inserted through the puncture hole 0.1 mm lateral of the spinal midline. The micropipette was lowered to ~0.4 mm beneath the spinal surface and 200 nL of AAV2retro was injected over 1 min. The micropipette was left in place for 30 s, raised to ~0.3 mm beneath the spinal surface, and another 200 nL of AAV2retro was injected over 1 min. After 30 additional seconds, the micropipette was slowly withdrawn and identical injections were performed on the other side of the spinal cord.

In the uninjured mice, identical injections targeting the same coordinates were performed in the intact C4 spinal cord: N = 4 mice received bilateral injection of intact gray matter, and N = 4 mice received bilateral injection of intact white matter. Following surgery, the muscles and skin were sutured and secured with surgical staples, and standard postoperative care was provided, including daily injections of lactated Ringer’s solution, banamine, and ampicillin for 3 days after surgery. Three weeks later, mice were sacrificed by transcardial perfusion with 0.01 PBS followed by perfusion with 4% paraformaldehyde in 0.1 M phosphate buffer. Brains and spinal cords were dissected and drop fixed in 4% paraformaldehyde for ~18 h at 4 °C, then cryoprotected by immersion in 30% sucrose at 4 °C.

### 2.3. Histology

The brain, cervical spinal cord, and lumbar spinal cord from each mouse were frozen and sectioned on a sliding microtome set to a thickness of 40 µm. Free-floating sagittal brain sections and transverse spinal cord sections were collected in series of 12. Brains were cut along the midline prior to sectioning, and the left and right sides from each mouse were independently sectioned and stained.

For immunofluorescent labeling, sections were washed in tris-buffered saline (TBS), blocked for 1 h at room temperature (RT) in 5% donkey serum and 0.25% Triton X-100 in TBS, and incubated overnight at 4 °C in blocking solution containing primary antibodies. Brain sections were incubated in 1 µg/mL polyclonal rabbit anti-mCherry (Encorbio RPCA-mCherry) and a 1:2000 dilution of polyclonal guinea pig anti-NeuN (Millipore Sigma abn90). Spinal cord sections were incubated in 1 µg/mL polyclonal chicken anti-RFP (Novus NBP1-97371), 0.5 µg/mL polyclonal rabbit anti-Iba1 (Cell Signaling 79394), and 1:1000 dilution of polyclonal guinea pig anti-NeuN (Millipore Sigma abn90). Sections were then washed in TBS and incubated for 2 h at RT in blocking solution containing secondary antibodies conjugated to Alexa Fluor (AF) fluorescent dyes (Jackson ImmunoResearch). Brain sections were incubated in 4 µg/mL of donkey anti-rabbit AF568 and donkey anti-guinea pig AF647. Spinal cord sections were incubated in 4 µg/mL of donkey anti-guinea pig AF488, donkey anti-chicken AF568, and donkey anti-rabbit AF647. Sections were then washed in TBS, mounted on gelatin-coated positively charged slides, and cover-slipped in Mowiol 4-88 mounting medium (Millipore Sigma 9002-89-5).

For Nissl counterstaining, spinal cord sections were mounted on gelatin-coated positively charged slides and dried overnight at 50 °C. Slides were immersed in 50% chloroform/50% ethanol for 30 min, then in 100% ethanol, 95% ethanol, 70% ethanol, and 50% ethanol for 1.5 min each. Slides were dipped in distilled water and immersed for ~15 s in 0.25% thionin buffered in 0.2 M acetic acid. Slides were then rinsed twice in distilled water and immersed in 50% ethanol, 70% ethanol, 95% ethanol, and 100% ethanol for 1.5 min each, immersed three times in isopropanol for 2 min, 5 min, and 5 min, and immersed three times in citrisolv clearing agent (Decon Labs) for 2 min, 5 min, and 5 min. Slides were then cover-slipped in DPX mounting medium (Millipore Sigma 06522).

Images were acquired using an Axio Scan Z1 slide-scanning microscope (Zeiss, Oberkochen, Germany) with 10× Plan-Apo objective (Zeiss 420640-9900-000) and Colibri 7 LED light source (Zeiss 423052-9770-000).

### 2.4. Analysis and Statistics: Brain

For both the left and right side of each brain, immunofluorescent images from one medial (0.9–1.2 mm from midline) and one lateral (1.7–2.0 mm from midline) sagittal brain section were exported and blinded. All sections were stained, scans acquired, and images exported in parallel under identical conditions. The same images were used for both cell count and fluorescence intensity analyses, but were separately blinded and analyzed as independent datasets. Blinding was performed by M.J.C. Analyses were performed by K.T.N., fully blinded to group identity. All raw data are included in the graphs: each dot represents an individual datapoint, and every acquired datapoint is shown. Effect sizes (Cohen’s *d*) were determined by comparing each group against injection of intact gray matter, and were calculated by dividing the difference between the group means by the pooled standard deviation. The 95% confidence interval (CI) provided for each effect size represents the CI of the difference between group means. Analyses were performed and graphs were generated using Graphpad Prism software version 6.0.

For cell count analysis, for each image the number of tdTomato-labeled neurons in the sensorimotor cortex was manually counted using Zeiss Zen 3.1 (blue edition) software by placing an event marker on each labeled neuron, then recording the total number of markers on each section. After counting, images were unblinded and statistical analysis was performed. Prior to statistical analysis, one mouse from group 1 (intact gray matter injection) and one mouse from group 2 (intact white matter injection) were excluded due to mistargeted injections based on spinal cord histology, resulting in N = 6–8 datapoints per group for both medial and lateral brain sections. Two statistical analyses were performed on these cell counts: one for medial brain sections, and one for lateral brain sections. In both cases, the number of tdTomato-labeled neurons in the sensorimotor cortex was compared among the four groups. Because sample sizes were not sufficiently large to determine whether the data were normally distributed, we used the non-parametric Kruskal–Wallis test. Both Kruskal–Wallis tests were statistically significant (*p* < 0.01). Each group was individually compared to each other group by performing Dunn’s post-tests (*p* values were adjusted to account for multiple comparisons).

Fluorescence intensity analysis was performed using FIJI (ImageJ) software version 2.16.0/1.54p [34]. First, a region of interest containing the sensorimotor cortex was cropped from each image. A binary mask was generated by duplicating the cropped image and running the Auto Threshold function (using the “default” method with “ignore black” enabled). The mean tdTomato fluorescence intensity was then obtained by redirecting the Measure function through the binary mask to the original image, resulting in specific quantification only of pixels that contained fluorescent signal. After quantification, images were unblinded and statistical analysis was performed. Two statistical analyses were performed on these fluorescence intensity measurements (excluding the two mice with mistargeted injections): one for medial brain sections, and one for lateral brain sections. As above, the mean fluorescence intensity was compared among the four groups by performing non-parametric Kruskal–Wallis tests. Both Kruskal–Wallis tests were statistically significant (*p* < 0.01). Each group was individually compared to each other group by performing Dunn’s post-tests (*p* values were adjusted to account for multiple comparisons).

### 2.5. Analysis and Statistics: Lumbar Spinal Cord

One immunolabeled L4 spinal cord section per mouse was imaged by confocal microscopy for quantification of tdTomato-labeled hindlimb corticospinal axons in the dorsal column white matter. For each section, a z-stack covering the entire dorsal column white matter was acquired using an Olympus BX63 FluoView FV3000 laser confocal microscope with UPlanFl 40×/1.30 oil immersion objective and Olympus FV31S-SW software (variable barrier filter with line averaging, 2048 × 2048 scan size, 1:1 aspect ratio, 0.54 µm section thickness, 120 µm aperture, 25% exposure, 400 V high voltage, 1× gain, 5% offset). As above, all sections were stained, scans acquired, and images exported in parallel under identical conditions. Z-stacks were blinded prior to analysis. Blinding was performed by M.J.C. Analysis was performed by K.T.N., fully blinded to group identity.

Analysis was performed using FIJI (ImageJ) software version 2.16.0/1.54p [34]. Each blinded z-stack was first converted to a maximum intensity projection by running the Z Project command and selecting Max Intensity as the projection type. A binary threshold was applied to the resulting image by running the Threshold function with the default settings (“default” method with “dark background” and “select range” enabled), the lower threshold level adjusted to 1497, and the upper threshold set to the maximum level. The unilateral left or right corticospinal tract was then selected and cropped (the left and right corticospinal tracts were independently analyzed as separate datapoints). The selected region was quantified by running the Analyze Particles function with the default settings (size of 0–infinity micron^2^ and circularity of 0.00–1.00), and the total number of particles (axons) was recorded. After quantification, images were unblinded and statistical analysis was performed. The number of tdTomato-labeled axons in the dorsal column white matter was compared among the four groups by performing a non-parametric Kruskal–Wallis test, which was statistically significant (*p* < 0.01). Each group was individually compared to each other group by performing Dunn’s post-tests (*p* values were adjusted to account for multiple comparisons).

## 3. Results

### 3.1. Experimental Design

The experimental design is summarized in Figure 1. Four groups of mice were compared in this study:1.Injection of AAV2retro into intact C4 gray matter;2.Injection of AAV2retro into intact C4 dorsal column white matter;3.Injection of AAV2retro into C4 gray matter bordering a C5 dorsal column lesion;4.Injection of AAV2retro into C4 dorsal column white matter bordering a C5 dorsal column lesion.

For all groups, each side of the spinal cord received 4 × 10^8^ vg of AAV2retro-tdTomato in 0.4 µL split evenly between two depths along a single injection track. Mice were sacrificed 3 weeks after treatment, and brains and spinal cords were analyzed by immunofluorescence and Nissl staining. In rodents, corticospinal neurons on one side of the brain almost entirely project to the contralateral side of the spinal cord [1,35]. Each spinal injection of AAV2retro thus separately targeted a distinct population of corticospinal neurons on the opposite side of the brain. Because we exclusively examined anatomical outcomes in this study, we analyzed each anatomically independent side of the brain or spinal cord as an independent datapoint. The results supported the validity of this approach: substantial differences between the left and right sides were observed in some mice, indicating that the two sides are anatomically distinct circuits that may be differently targeted by the separate AAV2retro injections on each side (see Section 3.3, below). Two mice were excluded due to mistargeted injections based on spinal cord histology (see Section 3.4, below), resulting in N = 6–8 datapoints per group.

### 3.2. Retrograde Transduction of Corticospinal Neurons in the Sensorimotor Cortex

In both medial (0.9–1.2 mm from midline) and lateral (1.7–2.0 mm from midline) sagittal brain sections, we analyzed the number of retrogradely transduced corticospinal neurons in the primary sensorimotor cortex by immunofluorescent staining of the tdTomato reporter gene and the neuronal marker NeuN (RBFOX3) (Figure 2). Across all groups, tdTomato-labeled neurons in the sensorimotor cortex were confined to layer 5, with no detectable labeling of cells in other cortical layers (Appendix A), consistent with specific targeting of layer 5 corticospinal projection neurons by retrograde transport.

Injection of AAV2retro into intact C4 gray matter resulted in limited retrograde transduction of corticospinal neurons that project to C4 spinal cord, which were largely restricted to the forelimb sensorimotor cortex (Figure 2A) [28,30]. In contrast, injection into C4 dorsal column white matter bordering a C5 dorsal column lesion resulted in significantly more tdTomato-labeled corticospinal neurons in both medial and lateral brain sections compared to injection of intact C4 gray matter, consistent with our hypothesis that this approach would increase transduction of corticospinal neurons. Injection into C4 spinal gray matter bordering a C5 dorsal column lesion produced variable results: extensive transduction of corticospinal neurons (similar to white matter injection) was observed in some brain sections, while more restricted transduction of the forelimb sensorimotor cortex (similar to intact gray matter injection) was observed in other sections (Figure 2). This could potentially reflect variable access to transected corticospinal neurons due to variable spread of AAV2retro from the gray matter injection site into the spinal cord lesion (see Section 3.4, below).

Remarkably, injection of AAV2retro into intact C4 dorsal column white matter resulted in the most extensive retrograde transduction of corticospinal neurons. Injection into intact white matter transduced significantly more corticospinal neurons in both medial and lateral brain sections than injection either into intact C4 gray matter or into gray matter bordering a C5 dorsal column lesion (Figure 2). Injection of AAV2retro into intact C4 white matter extensively transduced corticospinal neurons in the hindlimb sensorimotor cortex, which is located posterior of the forelimb sensorimotor cortex in medial brain sections (Figure 2A). This strongly suggests that AAV2retro can enter axons of passage and retrogradely transduce hindlimb corticospinal neurons after injection into the dorsal column white matter.

### 3.3. Quantification of Transduced Corticospinal Axons in the Lumbar Spinal Cord

We directly examined transduction of hindlimb corticospinal neurons by performing immunofluorescent labeling of the L4 lumbar spinal cord and quantifying the number of tdTomato-labeled corticospinal axons in the dorsal columns (Figure 3). After injection of AAV2retro into the intact C4 dorsal column white matter, large numbers of tdTomato-labeled hindlimb corticospinal axons were observed within the lumbar dorsal columns and innervating the lumbar gray matter (Figure 3B). In contrast, after injection of AAV2retro into the intact C4 gray matter, very few tdTomato-labeled axons were observed in the lumbar dorsal columns (Figure 3A). These rare labeled axons may originate from a small subpopulation of hindlimb corticospinal neurons that project collaterals to the injected C4 level of the cervical spinal cord [36,37] and thus could be targeted in limited numbers by injection of cervical gray matter. No labeled corticospinal axons were observed in the lumbar spinal cord in mice that received a C5 dorsal column lesion (Figure 3C,D), consistent with complete transection of the corticospinal tract.

As noted above, substantial differences between the left and right corticospinal tracts were observed in some mice, both in the number of tdTomato-labeled corticospinal axons in the dorsal columns and in the number of tdTomato-labeled corticospinal neurons in the sensorimotor cortex (Appendix A). This was observed even after AAV2retro injection into the intact dorsal column white matter, despite the white matter injections on the left and right sides being separated by only ~0.2 mm (Appendix A). This supports the analysis of each side as an anatomically independent circuit.

Together, the extensive labeling of layer 5 corticospinal neurons in the hindlimb motor cortex (Figure 2) and the large number of labeled axons in the lumbar corticospinal tract (Figure 3) provide strong evidence that AAV2retro enters axons of passage and retrogradely transduces hindlimb corticospinal neurons after injection into the cervical dorsal column white matter.

### 3.4. Accuracy and Safety of Spinal Cord Injections by Immunofluorescent and Nissl Staining

In the cervical spinal cord, we examined injection targeting by immunofluorescent labeling of tdTomato, and we examined tissue morphology and potential inflammation or damage by immunofluorescent labeling of the microglial marker Iba1 (Figure 4). Immunofluorescent labeling of tdTomato confirmed that most injections were accurately targeted to either the spinal gray matter or the dorsal column white matter. As noted above, injections were found to be mistargeted in one mouse from group 1 (intact gray matter injection) and one mouse from group 2 (intact white matter injection). The intended gray matter injections were overly deep and mistargeted the ventral white matter, while the intended white matter injections were overly lateral and mistargeted the dorsal gray matter (Appendix A). In both cases, transduction of corticospinal neurons was greatly reduced when compared to the other six datapoints from the same group (Appendix A). These two mice were excluded from further analysis. For two mice in group 3 (gray matter injection bordering SCI), injections were confirmed to be accurately targeted to the spinal gray matter (Figure 3C). For the other two mice in group 3, we did not find any spinal cord sections with an injection site labeled by tdTomato, possibly due to diffusion of the vector into the spinal cord lesion. Although we could not directly confirm that these injections were accurately targeted to spinal gray matter, counts of tdTomato-labeled corticospinal neurons in the corresponding brain sections were evenly distributed among the datapoints from the other two mice in this group (Figure 2), suggesting that AAV2retro was successfully delivered to the lesioned spinal cord. For example, the two mice from group 3 without detectable injection sites had the 8th, 5th, 4th, and 2nd most tdTomato-labeled neurons counted in lateral brain sections (Figure 2B). All injections in group 4 (white matter injection adjacent to SCI) were accurately targeted to the dorsal column white matter (Figure 3D).

We observed limited enrichment of Iba1 immunolabeling and minor disruption of tissue morphology by Nissl staining closely surrounding tracks of glass pipette passage through the spinal cord (Figure 4 and Figure 5). Limited inflammation surrounding injection tracks is typically observed after direct AAV injection into CNS tissue without detectable adverse effects [14,38,39]. In most mice that received a spinal cord injury, injection tracks were not detectable by Iba1 labeling or Nissl staining, likely because they were obscured by pathology from the C5 lesion site (Figure 5C,D). After SCI, enrichment of Iba1 immunolabeling and increased cellular density were observed surrounding the lesion site and spreading through the dorsal white matter multiple spinal levels above the C5 lesion (Figure 4C,D and Figure 5C,D).

In intact mice, identification of injection tracks by Nissl counterstaining further supported the accuracy of both gray and white matter injections (Figure 5A,B), although some white matter injection tracks appeared to touch the inner edge of the white matter (Figure 5B). Based on tdTomato immunolabeling, some white matter injections remained tightly confined within the dorsal column white matter (Figure 4D), while others appeared to spread a short distance into the adjacent gray matter (Figure 4B). In both cases, extensive retrograde transduction of corticospinal neurons was observed throughout both the forelimb and hindlimb sensorimotor cortical regions (Figure 2), suggesting that AAV2retro can diffuse through the white matter and broadly engage the corticospinal tract regardless of the exact targeting of the injection within the dorsal columns. In contrast, when injections were mistargeted to the ventral gray matter, transduction of corticospinal neurons was greatly reduced (Appendix A). This suggests that AAV2retro does not readily spread from the gray matter into the adjacent dorsal column white matter, and that positioning the needle tip within the dorsal white matter is essential.

Importantly, no inflammation or tissue damage was observed in sections adjacent to those containing an injection track by Nissl staining, suggesting that white matter injections did not cause substantial damage to the dorsal column white matter (Figure 5B). These results suggest that injections of AAV2retro into the intact dorsal columns were accurately targeted, did not cause meaningful damage beyond a narrow injection track, and resulted in extensive retrograde transduction of corticospinal neurons.

## 4. Discussion

We conclude that injection of AAV2retro into either the intact or transected dorsal column white matter retrogradely transduces corticospinal neurons throughout both the forelimb and hindlimb sensorimotor cortical regions. Although we hypothesized that transected axons would be susceptible to transduction by AAV2retro after SCI, we did not predict that AAV2retro would efficiently transduce axons of passage in the dorsal columns. Nonetheless, our results strongly suggest that AAV2retro can enter axons of passage in the intact spinal white matter.

Few previous studies have examined whether natural AAVs can enter axons of passage in the adult CNS. An early study reported transduction of dopaminergic neurons in the rat substantia nigra pars compacta after injecting AAV5 into the medial forebrain bundle, which connects limbic, midbrain, and hindbrain regions [40]. No transduction was observed when colchicine, an inhibitor of fast axonal transport, was infused into the medial forebrain bundle 15 min prior to AAV5 injection, suggesting that AAV5 reached the substantia nigra by axonal transport after entering axons of passage, rather than by diffusing along the white matter tract. Injection of natural AAVs into the rat or marmoset sciatic nerve also resulted in retrograde transduction of neurons in the spinal cord and dorsal root ganglion (DRG), which again was blocked by colchicine [41,42]. Both myelinated and unmyelinated DRG afferents were transduced by sciatic nerve injection [42]. This provides further evidence that natural AAVs can enter myelinated axons after direct injection into peripheral nerves or CNS white matter, but the underlying mechanisms are unknown. It is thus possible that AAV2retro enters axons of passage in the same manner as natural AAVs but is more efficiently transported retrograde to the cell body, resulting in extensive transduction of corticospinal neurons after injection into the dorsal column white matter. Alternatively, AAV2retro may engage a unique mechanism to enter myelinated axons more efficiently than other natural AAVs. It was previously reported that transient demyelination substantially enhanced retrograde transduction of central and peripheral neurons after injection of natural AAVs into the sciatic nerve, indicating that myelinated axons are resistant to transduction by natural AAVs [43]. It is unknown whether transient demyelination would similarly increase transduction of intact white matter by AAV2retro, but the strong retrograde transduction observed in this study at a low dose of only 4 × 10^8^ vg per side suggests that AAV2retro can efficiently enter myelinated corticospinal axons in the dorsal columns. It is also unknown whether AAV2retro can retrogradely transduce other populations of CNS neurons after injection into different CNS white matter tracts, or if this phenomenon is unique to the dorsal columns of the rodent spinal cord. These are important topics for future study and could improve the targeting of therapeutic gene delivery for motor and sensory disorders.

It is important to note that injections in this study were performed using a Picospritzer pressure injector [44]. This system uses controlled pressure pulses to dispense precise volumes from a fine glass micropipette: there is a linear relationship between the duration of the pressure pulse and the dispensed volume [44]. AAV infusion into the CNS is more commonly performed using a syringe pump, which applies continuous positive pressure to create convective flow and increase the spread of infusion [45,46,47]. However, a syringe pump is poorly suited for targeting superficial areas like the mouse dorsal columns, which are located only ~0.3–0.5 mm beneath the spinal surface, because continuous positive pressure is likely to produce reflux from the superficial injection site and leakage onto the spinal surface. It is also likely that continuous infusion using a syringe pump would increase the spread of AAV beyond the dorsal column white matter into the adjacent dorsal gray matter. In contrast, a picospritzer can deposit small volumes with minimal spread or reflux. In this study, we deposited 400 nL divided evenly between two depths along a single injection track targeting the dorsal columns on each side of the spinal cord. We did not observe leakage during any of the injections in this study, but after some white matter injections, we noted a small amount of reflux when the micropipette was removed from the spinal cord. This did not appear to reduce the efficacy of the six injections that accurately targeted the dorsal column white matter, as we observed consistently strong and broad transduction of corticospinal neurons in brain sections corresponding to all six injections (Figure 2).

Directly injecting AAV2retro into spinal white matter could provide several advantages for therapeutic gene delivery to corticospinal neurons. First, injection of the dorsal column white matter appears to transduce both forelimb and hindlimb corticospinal neurons more broadly than injection of spinal gray matter, which only targets corticospinal neurons that innervate the injected spinal level (Figure 2).

Second, white matter injection could increase specificity by avoiding uninjured cells in the spinal gray matter and restricting transduction to injured corticospinal neurons. Neuron-specific promoters such as the human synapsin 1 promoter can prevent mistargeting of glia in the spinal white matter [48]. In this study, after some white matter injections we observed limited spread of AAV2retro into the adjacent spinal gray matter. Off-target expression in spinal gray matter could potentially be addressed by several recently described enhancers that are specific for layer 5 cortical projection neurons in the mouse brain [21]. Although these enhancers have not yet been tested in the spinal cord and their activity in humans is unknown, it is possible that future studies will identify enhancers that eliminate off-target expression in the spinal cord and could completely restrict expression to corticospinal neurons after AAV2retro injection into the dorsal column white matter. Directly injecting AAV2retro into the corticospinal tract after SCI could thus precisely target regenerative gene therapy to only the injured neurons that must regenerate for voluntary motor function.

Third, although this approach requires direct injection of AAV2retro into the spinal cord, this could potentially be performed at the same time as surgical intervention to stabilize the spinal cord and treat the injury site. It is well established that without further intervention, regenerating corticospinal axons cannot efficiently traverse sites of spinal cord injury [49,50]. Therapies that promote regeneration of corticospinal neurons must be combined with additional treatment of the injury site to provide a permissive environment for corticospinal axon regeneration, such as implantation of a neural stem cell graft or bioengineered scaffold [50,51,52]. Although further preclinical studies are needed, it may be possible to implant a stem cell graft or scaffold into the injury site and inject AAV2retro rostral of the injury during the same surgical session: unlike direct brain injection, an additional invasive surgery would not be required.

Finally, this method could potentially target all upper motor neurons that project through the corticospinal tract, which in primates are found throughout the primary motor, premotor, cingulate, and somatosensory cortical regions [53,54]. Treating all of these regions by direct brain injection is unlikely to be feasible in human patients, and it remains uncertain whether enhancer sequences can restrict expression to such a precisely defined subpopulation of neurons after systemic delivery without compromising the strength of expression or exceeding the packaging capacity of AAV.

Injection of AAV2retro into the corticospinal tract is a promising method for therapeutic gene delivery to corticospinal neurons, but some challenges and uncertainties remain. In primates, descending corticospinal axons are not located in the dorsal columns. Rather, the primary (“lateral”) corticospinal tract that controls limb movement is found in the dorsolateral white matter, while a smaller (“anterior”) corticospinal tract that controls the axial muscles is found in the ventral white matter [55]. Further studies are needed to determine whether injection of AAV2retro into the primate corticospinal tract can efficiently target intact or transected corticospinal axons in primates as in mice. Treatment of SCI may also be improved by targeting additional spinal-projecting neurons in the midbrain and brainstem, which project through the rubrospinal, reticulospinal, olivospinal, and vestibulospinal tracts. It will be important to examine whether AAV2retro can efficiently target intact or transected axons in these descending extrapyramidal tracts in rodents and primates. In addition, although prior clinical trials have demonstrated that AAV vectors can be safely injected into the human CNS without adverse effects [14,15], and we observed no inflammation or tissue damage beyond a small injection track three weeks after intraspinal injection, further study of long-term outcomes after AAV2retro injection into the cervical white matter in preclinical models of spinal cord injury will be needed to comprehensively assess the safety of this method at longer intervals after treatment.

Preclinical development of regenerative gene therapies for SCI will also require improved methods for temporal control of gene expression. In two recent studies that examined functional outcomes in mice after SCI, the *PTEN* gene was deleted in corticospinal neurons by injection of AAV2retro into the spinal gray matter [30,56]. This treatment initially improved motor function, consistent with prior reports that *PTEN* deletion promotes extensive regeneration of corticospinal axons [9,30,56]. However, in both studies, side effects emerged over time, including hindlimb dystonia, excessive scratching, degeneration of corticospinal neurons, and a decline in motor function [30,56]. In another recent study of SCI in rats, overexpressing constitutively active Akt3 in corticospinal neurons promoted axon regeneration beyond the injury, but also caused dose-dependent behavioral seizures and a decline in motor function over time [8]. It thus appears that regenerative gene therapies must be transiently expressed in corticospinal neurons only until axon regeneration has restored the descending motor circuitry, and that prolonged upregulation of pro-growth signaling can cause toxicity and adverse side effects. One potential method for inducible gene expression in the CNS is the “Xon” system, which uses drug-induced alternative splicing of AAV mRNA to control protein translation [57]. This could provide temporal control of regeneration after gene delivery to corticospinal neurons, but this system has not yet been tested in human subjects. Ultimately, a clinically feasible method for temporal control of pro-growth signaling will be essential for safe and effective regenerative gene therapy in human SCI.

## 5. Conclusions

In this study, we initially hypothesized that AAV2retro injected into the dorsal column white matter adjacent to a spinal cord injury would spread to the rostral edge of the lesion and enter the transected ends of corticospinal axons. Although this approach was effective, such precise placement and timing of AAV2retro injection does not appear to be necessary: injection into the intact dorsal column white matter without an injury also effectively targeted both forelimb and hindlimb corticospinal neurons. This suggests that following a spinal cord injury, AAV2retro could be injected into the corticospinal tract distal from the injury site to avoid any adverse impact of the post-injury environment and prevent leakage of AAV into the lesion site. Indeed, compared to injection into the dorsal column white matter adjacent to a spinal cord injury, we found that injections into the intact dorsal columns were less variable and transduced on average more corticospinal neurons with significantly greater mean fluorescence intensity (Figure 2). This could potentially reflect variable loss of AAV due to leakage into the lesion site when injections were performed immediately rostral of a spinal cord injury, and suggests that injections distal from the site of injury might more consistently target gene delivery to corticospinal neurons. After SCI, transected corticospinal axons typically retract from the injury site and form terminal swellings known as retraction bulbs [58]. In mice, deletion of the *PTEN* gene one year after SCI can effectively promote corticospinal axon regeneration, indicating that chronically injured corticospinal neurons can still respond to regenerative gene therapies [59]. These findings suggest that regenerative gene delivery by injection of AAV2retro into the spinal white matter could potentially promote corticospinal axon regeneration even when injected distal from the lesion site in chronically injured subjects.

Injection of AAV2retro into spinal white matter could improve efficacy and specificity over existing methods for gene delivery to corticospinal neurons. Compared to direct injection of the primary motor cortex, this new approach would require much fewer invasive injections and could more extensively treat corticospinal neurons throughout the brain. Compared to direct injection of AAV2retro into spinal gray matter, injecting cervical white matter could more broadly treat both forelimb and hindlimb corticospinal neurons and reduce mistargeting of uninjured cells in spinal gray matter. Compared to infusion into the blood or cerebrospinal fluid, direct injection of AAV2retro into spinal white matter requires a much lower AAV dose and could more specifically target injured corticospinal neurons, decreasing the likelihood of off-target side effects.

While further preclinical research is needed to develop safer gene therapies for controlled corticospinal axon regeneration in human SCI, direct injection of AAV2retro into spinal white matter could potentially provide a highly effective method for the targeted delivery of these emerging therapeutics to corticospinal neurons.

## Figures and Tables

**Figure 1 brainsci-15-01058-f001:**
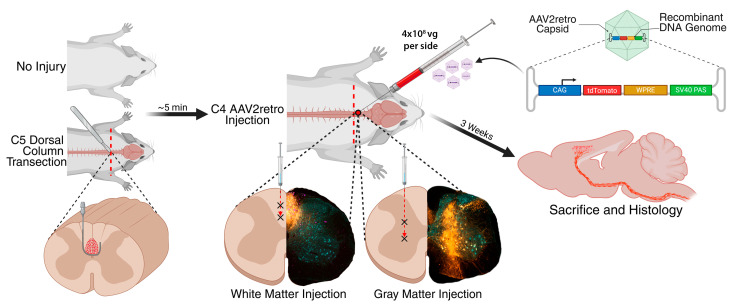
Experimental Design. Sites of AAV2retro injection are indicated by an “X” symbol. CAG: chicken beta-actin gene promoter. WPRE: woodchuck hepatitis virus post-transcriptional regulatory element. SV40 PAS: simian virus 40 polyadenylation signal. Vg: vector genomes. Partially created in BioRender. Castle, M. (2025). https://BioRender.com/0dodf9o (accessed on 27 September 2025).

**Figure 2 brainsci-15-01058-f002:**
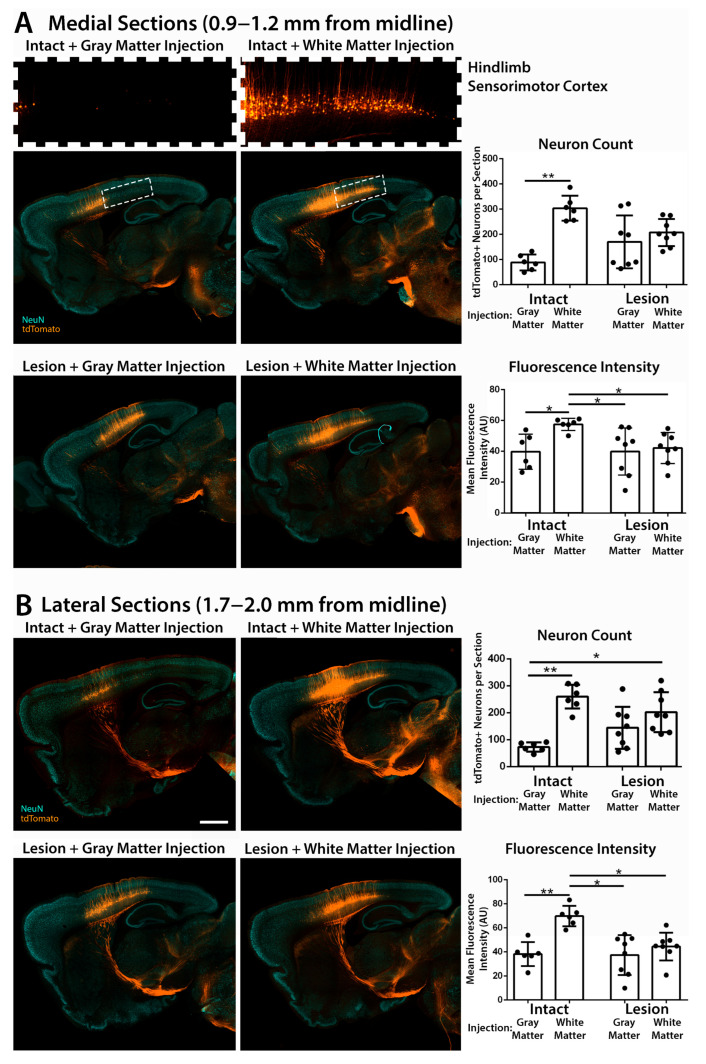
Injection of AAV2retro into dorsal column white matter extensively transduces corticospinal neurons in both the forelimb and hindlimb sensorimotor cortical regions. Medial (**A**) and lateral (**B**) sensorimotor cortex were analyzed by immunolabeling of tdTomato (orange) and NeuN (blue) in 40 µm sagittal brain sections. High-magnification images of the hindlimb sensorimotor cortex (location indicated by a box on the corresponding low-magnification image) demonstrate that injection of intact white matter transduces hindlimb corticospinal neurons, while injection of intact gray matter does not (**A**). The number of tdTomato-labeled neurons and the mean intensity of tdTomato fluorescence were compared among groups. Compared to injection of AAV2retro into intact gray matter, injection of intact white matter increased the number of tdTomato-labeled neurons by an average of 215.2 per medial brain section, with an effect size (Cohen’s *d*) = 5.15, 95% CI [161.4, 269.0], and by 185.2 per lateral section (*d* = 5.16 [139.0, 231.3]). Injection of intact white matter also increased the mean intensity of tdTomato fluorescence in medial (*d* = 2.09 [6.8, 28.7]) and lateral (*d* = 3.42 [19.8, 43.5]) sections. White matter injection adjacent to SCI increased the number of tdTomato-labeled neurons by an average of 118.6 per medial section (*d* = 2.58 [64.5, 172.7]) and 129.3 per lateral section (*d* = 2.24 [61.5, 197.1]), but did not substantially change the mean fluorescence intensity in medial (*d* = 0.23 [−10.1, 14.9]) or lateral (*d* = 0.58 [−6.54, 19.2]) sections. Gray matter injection adjacent to SCI increased the number of tdTomato-labeled neurons by an average of 81.6 per medial section (*d* = 0.98 [−15.9, 179.2]) and 71.2 per lateral section (*d* = 1.18 [−0.15, 142.2]), but did not change the mean fluorescence intensity in medial (*d* = 0.0077 [−16.1, 16.3]) or lateral (*d* = −0.057 [−17.5, 15.8]) sections. All sections were stained and imaged in parallel under identical conditions. Adjustments to improve visibility were applied identically to all images. Scale bars: 1 mm. Error bars represent standard deviation. Statistical significance was determined by performing Kruskal–Wallis tests with Dunn’s post-tests. All statistically significant differences between groups are shown on the graphs. *: *p* < 0.05; **: *p* < 0.01. AU: arbitrary units.

**Figure 3 brainsci-15-01058-f003:**
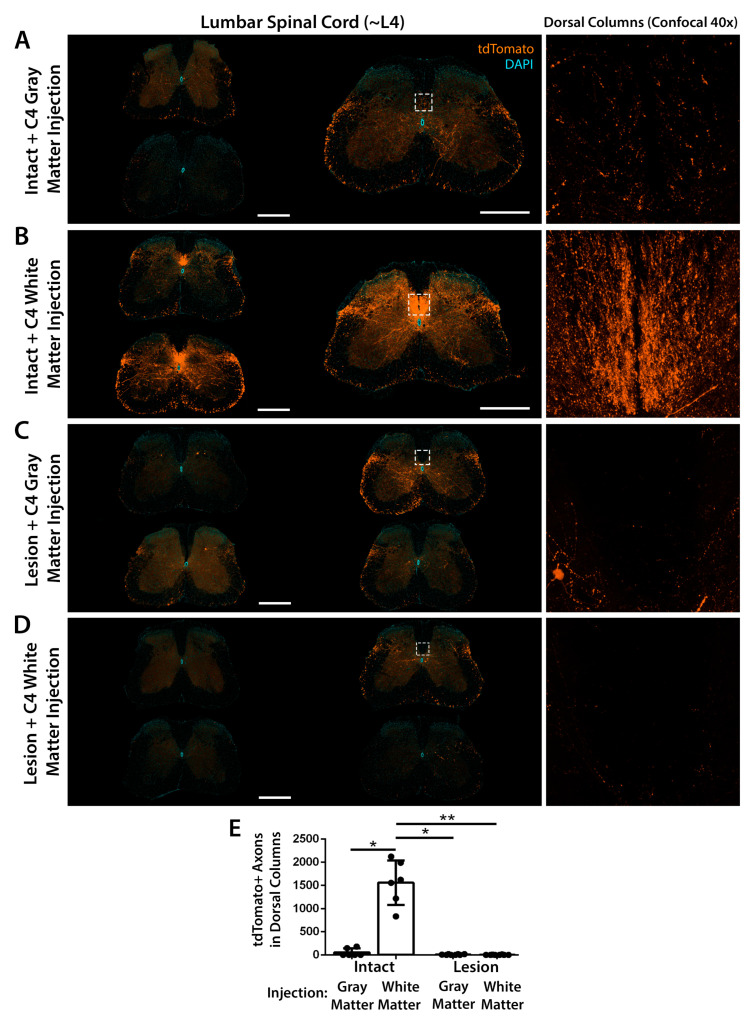
Hindlimb corticospinal axons in the lumbar spinal cord are strongly transduced by injection of AAV2retro into the cervical dorsal column white matter. The number of corticospinal axons in the lumbar (L4) spinal cord was quantified by immunolabeling of tdTomato (orange) and confocal microscopy of the dorsal column white matter. Each low-magnification image is from a different mouse, and all analyzed spinal cord sections are shown. A representative confocal image of the dorsal columns from each group is also shown (location indicated by a box on the corresponding low-magnification image). Compared to injection of AAV2retro into intact C4 gray matter (**A**), injection into intact C4 white matter (**B**) increased the number of tdTomato-labeled corticospinal axons in the dorsal columns by an average of 1501 per section (*d* = 4.36 [1058, 1944]). No corticospinal axons were detected in the dorsal columns of mice that received a C5 dorsal column lesion (**C**,**D**), consistent with complete transection of the corticospinal tract. All sections were stained and imaged in parallel under identical conditions. Adjustments to improve visibility were applied identically to all images. Scale bars: 0.5 mm. Error bars represent standard deviation. Statistical significance was determined by performing Kruskal–Wallis tests with Dunn’s post-tests (**E**). All statistically significant differences between groups are shown on the graphs. *: *p* < 0.05; **: *p* < 0.01.

**Figure 4 brainsci-15-01058-f004:**
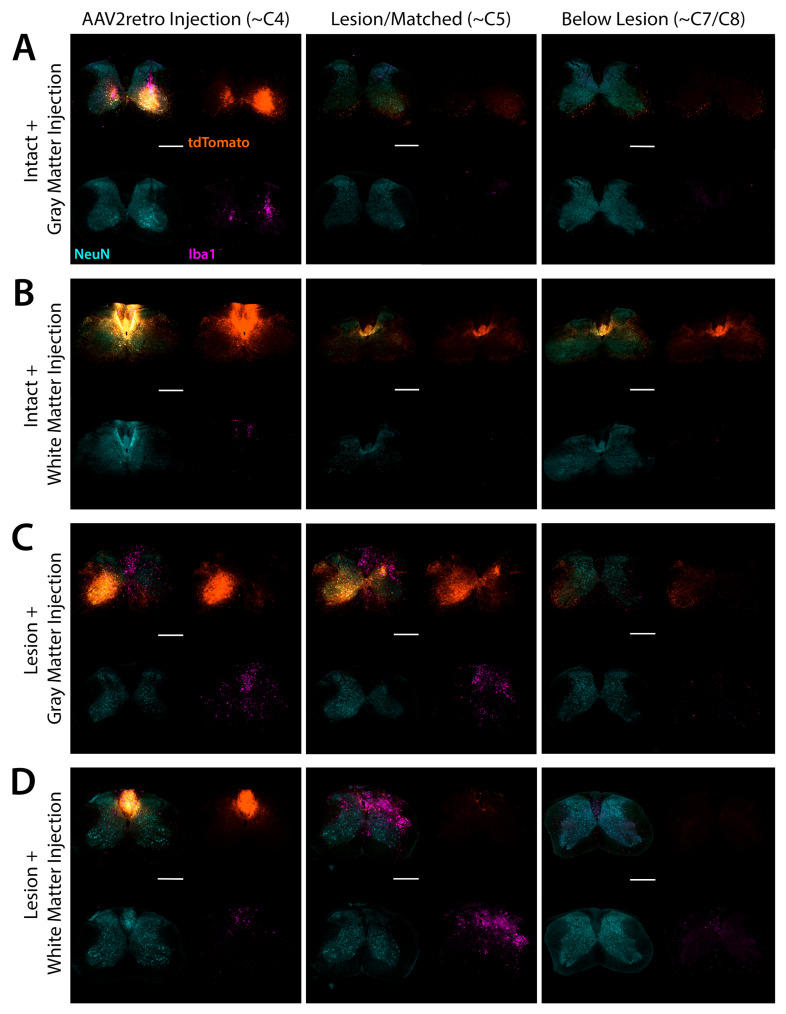
AAV2retro injections in the cervical spinal cord were accurately targeted and did not provoke microglial inflammation. Immunofluorescent labeling of tdTomato (orange), NeuN (blue), and Iba1 (magenta) in 40 µm transverse spinal cord sections indicates that AAV2retro injections were accurately targeted to the spinal gray matter (**A**,**C**) or dorsal column white matter (**B**,**D**). All images in each row are from the same mouse. All sections were stained and imaged in parallel under identical conditions, and adjustments to improve visibility were applied identically to all images. Scale bar: 0.5 mm.

**Figure 5 brainsci-15-01058-f005:**
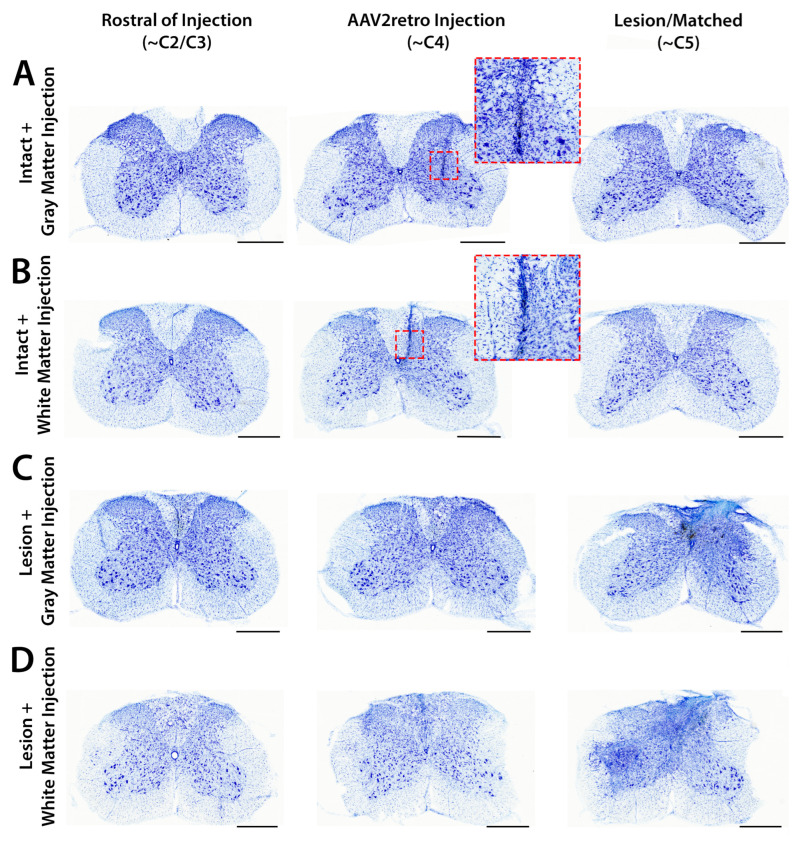
Nissl staining suggests that AAV2retro injection did not cause tissue damage or inflammation. After Nissl staining of 40 µm transverse spinal cord sections, increased cellular density was tightly confined to AAV2retro injection tracks in gray or white matter, with no detectable pathology beyond the injection track or in adjacent sections (**A**,**B**). Insets show injection tracks at higher magnification. After SCI, increased cellular density reflects inflammation and tissue damage surrounding the lesion site and extending through the dorsal column white matter multiple spinal levels above the injury (**C**,**D**). All images in each row are from the same mouse. All sections were stained and imaged in parallel under identical conditions, and adjustments to improve visibility were applied identically to all images. Scale bar: 0.5 mm.

## Data Availability

The original contributions presented in this study are included in the article and Appendix A. Further inquiries can be directed to the corresponding author.

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
