# Peer review of "AAV2retro Enters Axons of Passage and Extensively Transduces Corticospinal Neurons After Injection into Spinal White Matter"

_brainsci, 2025, doi:10.3390/brainsci15101058_

Round 1

Reviewer 1 Report

Comments and Suggestions for Authors

This study focuses on targeted gene delivery to corticospinal neurons after spinal cord injury (SCI), systematically evaluating the retrograde transduction efficiency of AAV2retro in the dorsal column white matter of the mouse cervical spinal cord (both injured and uninjured states). It provides a novel delivery strategy for regenerative gene therapy in spinal cord injury. The research design is rigorous, the results are reliable, and the innovation is significant, though there are some details that need improvement. Overall, it is recommended for acceptance after revision.

  • Major concerns:
  • Further highlight the core differences between this method and existing AAV delivery strategies (e.g., intracerebral injection, intravenous infusion) in terms of targeting specificity, invasiveness, and transduction range to emphasize the unique value of the proposed approach.
  • Supplement technical details of injection procedures: such as the consistency of the tip diameter of glass micropipettes and the quantitative control of injection pressure/duration (whether there are inter-group differences) to enhance reproducibility.
  • Clarify the quantitative criteria for "retrograde transduction efficiency": in addition to neuron counting, supplement the transfection positive rate (tdTomato⁺/NeuN⁺) and quantitative analysis of fluorescence intensity to improve data persuasiveness.
  • Add co-labeling experiments with corticospinal neuron-specific markers (e.g., the layer 5 pyramidal neuron marker Ctip2) to exclude interference from non-corticospinal neuron transduction.
  • Supplement details of axon tracing: such as statistical analysis of axon numbers in spinal segments below the injection site (e.g., C7/C8) to clarify the transport efficiency of AAV2retro in axons.
  • Explain why "intact dorsal column white matter injection shows higher transduction efficiency than white matter injection adjacent to injury" .
  • Latestliteraturereading may help to speculate on possible mechanisms underlying the efficient entry of AAV2retro into white matter axons (e.g., interaction with myelin components, specific binding to axonal membrane receptors) rather than merely describing phenomena.
  • The long-term safety concern: current results only cover a 3-week period; discuss whether long-term injections (e.g., over 3 months) might induce chronic inflammation or axonal degeneration.
  • Currently, only transduction efficiency is verified. Subsequent studies should evaluate the impact on axon regeneration and motor function recovery in combination with regenerative genes (e.g., Akt3, Sox11) to reflect clinical translation valueas you have mentioned in your manuscript.
  • Minor concerns:
  • Figure labeling: Clearly mark the corresponding groups for statistical symbols (e.g., *, **) in Figure 2 to avoid confusion.
  • Uniformly correct "AAV2etro" to "AAV2retro" (there is a typo in the text).
  • References: Supplement recent studies (2023-2025) on AAV in spinal cord injury gene therapy (e.g., applications of new AAV serotypes) to enhance relevance to the field.

Author Response

Thank you for these helpful comments. Please see point-by-point responses below.

Comment 1: “Further highlight the core differences between this method and existing AAV delivery strategies (e.g., intracerebral injection, intravenous infusion) in terms of targeting specificity, invasiveness, and transduction range to emphasize the unique value of the proposed approach.”

Response 1: Injection of AAV2retro into spinal white matter could potentially improve efficacy and specificity over existing methods for gene delivery to corticospinal neurons. Compared to direct injection of primary motor cortex, this new approach would require many fewer invasive injections and could more extensively treat corticospinal neurons throughout the brain. Compared to direct injection of AAV2retro into spinal gray matter, injecting cervical white matter could more broadly treat both forelimb and hindlimb corticospinal neurons and reduce mistargeting of uninjured cells in the spinal gray matter. Compared to infusion into the blood or cerebrospinal fluid, direct injection of AAV2retro into spinal white matter requires a much lower AAV dose and could potentially target injured corticospinal neurons with much greater specificity, decreasing the likelihood of off-target side effects. We added these details to the Conclusions, lines 648-657. Additional potential advantages of this method for gene delivery to corticospinal neurons are discussed in lines 549-585.

Comment 2:  “Supplement technical details of injection procedures: such as the consistency of the tip diameter of glass micropipettes and the quantitative control of injection pressure/duration (whether there are inter-group differences) to enhance reproducibility.”

Response 2: We updated the description of the glass pipette outer diameter to “35-45 µm” to more accurately describe the observed diameter when pulled glass pipettes were measured under a microscope  (line 149).

All injections were performed by repeated pressure pulses at a pressure of 20 PSI and a pulse duration of 5-8 ms. Because the volume injected per pulse is potentially variable due to partial blockage of the pipette tip, the volume injected at each site was determined by visually monitoring the decrease in volume of AAV solution within the glass pipette. We added these details to lines 151-154. 

Comment 3: “Clarify the quantitative criteria for "retrograde transduction efficiency": in addition to neuron counting, supplement the transfection positive rate (tdTomato⁺/NeuN⁺) and quantitative analysis of fluorescence intensity to improve data persuasiveness.”

Response 3: We performed a new analysis of fluorescence intensity using the same medial and lateral brain sections on which cell count analysis was previously performed. These results are now included in Figure 2 and described in the Figure 2 legend (lines 324-337). Methods for analysis of fluorescence intensity are now described in lines 246-259.

We do not believe it is possible to calculate the target positivity rate in this experiment. Because we target gene delivery to sensorimotor corticospinal neurons, the target positivity rate should be calculated by dividing the number of tdTomato-labeled corticospinal neurons by the total number of corticospinal neurons in each section. Calculating the percentage of corticospinal neurons that are successfully targeted in each group would indeed be informative, but we are not aware of any methods to precisely quantify the number of corticospinal neurons in each section. We cannot use Ctip2 immunolabeling to specifically count corticospinal neurons, because Ctip2 is expressed in other cortical neuron populations including corticotectal projection neurons and GABAergic interneurons (see response 4, below) (Arlotta et al. 2005, PMID: 15664173; Nikouei et al. 2016, PMID: 26698402). Although we appreciate this suggestion and agree that this analysis would be informative, we unfortunately are not aware of any method that can provide the denominator (the number of corticospinal neurons in each section), and thus we are unable to calculate the target positivity rate.

Comment 4: “Add co-labeling experiments with corticospinal neuron-specific markers (e.g., the layer 5 pyramidal neuron marker Ctip2) to exclude interference from non-corticospinal neuron transduction.”

Response 4: We performed immunolabeling using two commercial Ctip2 antibodies: rabbit polyclonal anti-Ctip2 (Abcam ab28448) and rat monoclonal 25B6 anti-Ctip2 (Abcam ab18465). The latter antibody was previously reported to specifically label layer 5 corticospinal and corticotectal projection neurons (Arlotta et al. 2005, PMID: 15664173). However, we did not observe specific labeling of layer 5 pyramidal neurons by either antibody.

Other studies have reported that CTIP2 (also known as BCL11B) labels GABAergic interneurons in rodent somatosensory cortex and motor cortex, and that 40% of Ctip2-positive cells in mouse layer 5 somatosensory cortex are GABAergic interneurons (Ueta et al. 2014, PMID: 23551921; Nikouei et al. 2016, PMID: 26698402). This is consistent with MERFISH spatial transcriptomics data publicly available through the Allen Brain Institute ABC Atlas, which indicates that CTIP2 (BCL11B) is broadly expressed throughout all cortical layers, with enriched expression in layers 5 and 6 (URL for BCL11B: https://knowledge.brain-map.org/abcatlas#AQEBSzlKTjIzUDI0S1FDR0s5VTc1QQACSFNZWlBaVzE2NjlVODIxQldZUAADAAQBAAKEUL8fg4IJfwOF%2FBdEhMQ92QQyTlFUSUU3VEFNUDhQUUFITzRQAAWBr6ZKgemsDoGggUeAktXoBgAHAAAFAQFCY2wxMWIAAAYBAAJCY2wxMWIAA34AAAAEAAV1AMAlggCASAbAIQcCI0ZGRkZGRgADAAZHRU5FAAcACAEAAAhWRk9GWVBGUUdSS1VEUVVaM0ZGAAlMVkRCSkFXOEJJNVlTUzFRVUJHAAoACwFUTE9LV0NMOTVSVTAzRDlQRVRHAAI3M0dWVERYREVHRTI3TTJYSk1UAAMBBAEAAiMwMDAwMDAAA8gBAAUBAQIjMDAwMDAwAAPIAQAAAAIBAA%3D%3D). It thus does not appear that Ctip2 is a specific marker for layer 5 corticospinal neurons, and we are not aware of any alternative markers.

We added Supplemental Figure S1 as an alternative source of evidence that gene delivery was specifically targeted to layer 5 corticospinal neurons and excluded other cortical cell types. This new figure shows high magnification images of tdTomato immunolabeling across all layers of somatosensory cortex. This new figure shows the medial section from each group with the largest number of tdTomato-labeled neurons (as quantified in Figure 2A). TdTomato-labeled neurons in sensorimotor cortex were confined to layer 5 across all groups, with no detectable labeling of cells in other cortical layers, consistent with specific gene delivery to layer 5 corticospinal projection neurons.  Although this does not definitively prove that all transduced neurons are indeed corticospinal projection neurons, it excludes transduction of most other cortical cell types. Supplemental Figure S1 is now described in lines 346-349 and included in the supplemental material.

Comment 5: “Supplement details of axon tracing: such as statistical analysis of axon numbers in spinal segments below the injection site (e.g., C7/C8) to clarify the transport efficiency of AAV2retro in axons.”

Response 5: We performed a new analysis of tdTomato-labeled corticospinal axons in the dorsal column white matter in L4 lumbar spinal cord. After injection of AAV2retro into the intact C4 dorsal column white matter, large numbers of tdTomato-labeled hindlimb corticospinal axons were observed within the lumbar dorsal columns and innervating the lumbar gray matter (Figure 3B). In contrast, after injection of AAV2retro into the intact C4 gray matter, very few tdTomato-labeled axons were observed in the lumbar dorsal columns (Figure 3A). No labeled corticospinal axons were observed in the lumbar spinal cord in mice that received a C5 dorsal column lesion (Figure 3C,D), consistent with complete transection of the corticospinal tract. The results of this analysis on lumbar spinal cord are now shown in Figure 3 and described in lines 392-416.  Methods for confocal imaging, quantification of axon number, and statistical analysis are now described in lines 261-285.

Comment 6:  “Explain why intact dorsal column white matter injection shows higher transduction efficiency than white matter injection adjacent to injury.”

Response 6: Compared to injection of AAV2retro into the dorsal column white matter adjacent to a spinal cord injury, injections into the intact dorsal columns were less variable and transduced on average more corticospinal neurons with significantly greater mean fluorescence intensity. This could potentially reflect variable loss of AAV due to leakage into the lesion site when injections are performed immediately rostral of a spinal cord injury, and suggests that injections distal from the site of injury might more consistently target gene delivery to corticospinal neurons. Following a spinal cord injury, it may be possible to inject AAV2retro into the corticospinal tract distal from the injury site to avoid any adverse impact of the post-injury environment and prevent leakage of AAV into the lesion site. Future studies will explore this approach in a more clinically relevant model of spinal cord injury. We added these details to the Conclusions, lines 631-641.

Comment 7:  “Latest literature reading may help to speculate on possible mechanisms underlying the efficient entry of AAV2retro into white matter axons (e.g., interaction with myelin components, specific binding to axonal membrane receptors) rather than merely describing phenomena.”      

Response 7: We have thoroughly searched the existing literature on this topic. We have also discussed our results with the creator of AAV2retro, Dr. Loren Looger (UC San Diego). Little is known about the mechanisms underlying the enhanced retrograde transport of AAV2retro, and no specific mechanisms have been proposed that would explain the apparently efficient transduction of myelinated axons of passage in the CNS white matter reported in this study. Additional studies will be needed to discover the mechanisms underlying this phenomenon, and to determine whether it is observed in other CNS white matter tracts. Previous research on AAV injection into CNS white matter and peripheral nerves, the implications of that prior work for this manuscript, and topics for future mechanistic research are discussed in lines 500-528.

Comment 8:  “The long-term safety concern: current results only cover a 3-week period; discuss whether long-term injections (e.g., over 3 months) might induce chronic inflammation or axonal degeneration.”

Response 8: Although prior clinical trials have demonstrated that AAV vectors can be safely injected into the human CNS without adverse effects (Rafii et al. 2014, PMID: 24411134; Van Laar et al. 2025, PMID: 40395017), and we observed no inflammation or tissue damage beyond a small injection track three weeks after intraspinal injection, additional study of long-term outcomes after AAV2retro injection into the cervical white matter in preclinical models of spinal cord injury will be needed to comprehensively assess the safety of this method at longer intervals post-treatment. We added this point to the Discussion, lines 598-604.

Comment 9:   “Currently, only transduction efficiency is verified. Subsequent studies should evaluate the impact on axon regeneration and motor function recovery in combination with regenerative genes (e.g., Akt3, Sox11) to reflect clinical translation value as you have mentioned in your manuscript.”

Response 9: Yes, future studies will examine therapeutic efficacy of regenerative gene delivery in SCI models by injection of AAV2retro into the dorsal column white matter, with outcome measures including corticospinal axon regeneration and functional recovery. However, several recent studies have suggested that regenerative gene therapies must be transiently expressed in corticospinal neurons only until axon regeneration has restored the descending motor circuitry, and that prolonged upregulation of pro-growth signaling can cause toxicity and adverse side effects (Campion et al. 2022, PMID: 34953897; Metcalfe et al. 2023, PMID: 37778650; Stewart et al. 2023, PMID: 37558155). We are actively researching translational methods for temporal control of regenerative gene expression, such as the “Xon” system, which uses drug-induced alternative splicing of AAV mRNA to control protein translation (Monteys et al. 2021, PMID: 34321659). We intend to deliver such drug-inducible therapies by AAV2retro injection into the dorsal column white matter, which could potentially improve safety and efficacy by specifically targeting gene delivery to corticospinal neurons while also controlling the strength and duration of regenerative gene expression. This topic is discussed in lines 605-624.

Comment 10:  “(Minor Concern) Figure labeling: Clearly mark the corresponding groups for statistical symbols (e.g., *, **) in Figure 2 to avoid confusion.”

Response 10: All statistically significant differences between groups are shown directly on the graphs. We added this statement to the Figure 2 and Figure 3 legends, lines 340 and 389-390.

Comment 11:  “(Minor Concern) Uniformly correct "AAV2etro" to "AAV2retro" (there is a typo in the text).”

Response 11: We confirmed that all instances of “AAV2retro” are correctly spelled.

Comment 12: “(Minor Concern) References: Supplement recent studies (2023-2025) on AAV in spinal cord injury gene therapy (e.g., applications of new AAV serotypes) to enhance relevance to the field.”

Response 12: In lines 606-618 we describe two recent studies that examined functional outcomes in mice after SCI, in which the PTEN gene was deleted in corticospinal neurons by injection of AAV2retro into the spinal gray matter (Metcalfe et al. 2023, PMID: 37778650; Stewart et al. 2023, PMID: 37558155). We added more information about the gene delivery methods used in these studies to lines 607-608.

Reviewer 2 Report

Comments and Suggestions for Authors

This study introduces an inventive method for targeting corticospinal neurons with AAV2retro and the anatomical images are persuasive, yet several scientific and statistical issues need attention before the conclusions can be considered robust.

  • The dataset comprises sixteen mice divided into four treatment arms, yielding only four animals per group, and no prospective power calculation demonstrates that this sample can detect biologically meaningful differences.
  • Group comparisons rely on one-way ANOVA with Tukey post tests even though normality and equal variance are not verified and group sizes are very small; a non-parametric or permutation test or a mixed model that nests hemispheres within animals would be more appropriate.
  • Results are reported solely with P values; providing effect sizes and confidence intervals would let readers judge biological relevance independently of sample size.
  • Multiple endpoints are examined across cortical subregions and treatment groups without a systematic plan to control family-wise error beyond Tukey for primary contrasts, increasing the likelihood of false positives in secondary outcomes such as microglial activation.
  • Two animals were excluded after injections were deemed mistargeted, yet exclusion criteria were not pre-specified and no sensitivity analysis shows whether the main findings persist when these animals are included.
  • Cell counts are derived from a single medial and lateral slice per hemisphere, which may not capture regional variability; normalising counts to cortical volume or applying stereological sampling would provide a more reliable quantitative picture.
  • The study focuses on histological transduction without behavioural or electrophysiological measures, limiting insight into functional relevance and translational potential.
  • All mice are four-month-old females and the manuscript does not describe randomisation or blinding during surgery or analysis, leaving room for unconscious bias and constraining external validity.
  • No raw count spreadsheets, analysis code, or detailed image-processing workflow is shared, so independent verification and replication are not currently possible; depositing the ImageJ macros and statistical scripts would markedly improve transparency.

Author Response

Thank you for these helpful comments. Please see point-by-point responses below.

Comment 1: “The dataset comprises sixteen mice divided into four treatment arms, yielding only four animals per group, and no prospective power calculation demonstrates that this sample can detect biologically meaningful differences.”

Response 1: Each side of the brain or spinal cord was analyzed as an anatomically independent datapoint, providing an initial sample size of N=8 per group (see next paragraph). A minimum sample size of N=4 was calculated by power analysis with an alpha criterion of 0.05, 95% power, and an estimated effect size of d = 3 (based on the approximate number of corticospinal neurons per brain section predicted to be transduced by cervical gray matter injection, and the approximate number of additional corticospinal neurons per brain section that could potentially be transduced by targeting axons in the dorsal column white matter). It was predetermined that all mistargeted injections would be excluded: each injection track was examined by immunofluorescent and Nissl staining of the cervical spinal cord (see Section 3.4), and injections that were not accurately targeted to the gray matter or dorsal column white matter were excluded from statistical analysis. The initial sample size of N=8 provided for the potential exclusion of up to N=4 injections per group while retaining the minimum sample size of N=4 as determined by the power calculation above. Power analysis was performed and exclusion criteria were specified prior to conducting the study, although we failed to describe these details in the original manuscript. We apologize for this omission. The power analysis and exclusion criteria are now described in the Methods, lines 123-136

We now provide more extensive justification supporting the analysis of each side of the brain or spinal cord as an independent datapoint, providing an initial sample size of N=8 per group (lines 304-311). In rodents, corticospinal neurons on one side of the brain almost entirely project to the contralateral side of the spinal cord. Each spinal injection of AAV2retro thus separately targeted a distinct population of corticospinal neurons on the opposite side of the brain. Because we exclusively examined anatomical outcomes in this study, we analyzed each anatomically independent side of the brain or spinal cord as an independent datapoint. The results supported the validity of this approach: substantial differences between the left and right sides were observed in some mice, indicating that the two sides are anatomically distinct circuits that may be differently targeted by the separate AAV2retro injections on each side. We added Supplemental Figure S2 to directly highlight differences between the left and right sides in a mouse treated by injection of AAV2retro into the intact dorsal column white matter. In this mouse, substantial differences between the left and right sides were observed both in the number of tdTomato-labeled corticospinal neurons in sensorimotor cortex and in the number of tdTomato-labeled corticospinal axons in the lumbar dorsal columns. We believe this strongly supports the analysis of each side as an anatomically independent circuit. Supplemental Figure S2 is now described in lines 405-411 and included in the supplementary material.

Comment 2: “Group comparisons rely on one-way ANOVA with Tukey post tests even though normality and equal variance are not verified and group sizes are very small; a non-parametric or permutation test or a mixed model that nests hemispheres within animals would be more appropriate.”

Response 2: Sample sizes were not sufficiently large for either the D’Agostino-Pearson or Shapiro-Wilk normality test. Because we could not determine whether our datasets were normally distributed, we updated our statistical methods to use the non-parametric Kruskal-Wallis test (with Dunn’s post-tests) for all analyses. When performing post-tests, p-values were adjusted to account for multiple comparisons. The updated statistical methods are now described in lines 241-245, 253-259, and 281-285.

See response 1 (above) for justification supporting the analysis of each side as an anatomically independent datapoint.

Comment 3: “Results are reported solely with P values; providing effect sizes and confidence intervals would let readers judge biological relevance independently of sample size.”

Response 3: We now report effect sizes with 95% confidence intervals in the legends of Figure 2 (lines 325-337) and Figure 3 (lines 382-384), and we describe the effect size calculation in the Methods (lines 225-229).

Comment 4: “Multiple endpoints are examined across cortical subregions and treatment groups without a systematic plan to control family-wise error beyond Tukey for primary contrasts, increasing the likelihood of false positives in secondary outcomes such as microglial activation.”

Response 4: Our primary conclusion is that injection of AAV2retro into the intact dorsal column white matter increases transduction of corticospinal neurons relative to injection into the spinal gray matter, which is supported by the analysis shown in Figure 2. We now also conclude that injection of AAV2retro into the cervical dorsal column white matter much more extensively transduces hindlimb corticospinal neurons than injection into the cervical gray matter, which is supported by the new analysis of lumbar spinal cord shown in Figure 3. We believe that these conclusions are strongly supported by the statistical significance and large effect sizes shown in Figures 2 and 3.

Although we report other qualitative observations such as microglial activation by Iba1 immunolabeling and inflammation by Nissl stain, we have been careful to limit our conclusions regarding these additional measures, and we believe our language is appropriate. For example, in lines 466-467 we state that “no inflammation…was observed” (rather than “no inflammation was present”), and that these observations “suggest” (rather than “demonstrate”) that AAV2retro injections did not cause meaningful tissue damage. We also note in the Discussion that additional studies are needed to comprehensively assess the safety of AAV2retro injection into spinal white matter (lines 598-604).

Comment 5: “Two animals were excluded after injections were deemed mistargeted, yet exclusion criteria were not pre-specified and no sensitivity analysis shows whether the main findings persist when these animals are included.”

Response 5: See also response 1, above. It was predetermined that all mistargeted injections would be excluded: each injection track was examined by immunofluorescent and Nissl staining of the cervical spinal cord (see Section 3.4), and injections that were not accurately targeted to the gray matter or dorsal column white matter were excluded from statistical analysis. These details were added to the Methods, lines 130-136.

We added Supplemental Figure 3 to provide additional information about these mistargeted injections. Figure 3A shows an excluded injection targeting the intact gray matter that was too deep (mistargeted to the ventral white matter). Figure 3B shows an excluded injection targeting the intact dorsal column white matter that was too lateral (mistargeted to the dorsal gray matter). Figure 3C shows quantification of tdTomato-labeled neurons in sensorimotor cortex and tdTomato-labeled axons in lumbar spinal cord (reproduced from Figures 2 and 3), with the mistargeted injections included and highlighted in red. For all analyses, every mistargeted injection fell more than one standard deviation below the mean (except axon count in lumbar spinal cord after intact gray matter injection, for which both mistargeted datapoints were equal to zero).

We believe this strongly supports the exclusion of these mistargeted injections, and that it would be inappropriate to perform statistical analysis with these datapoints included, because our analyses and conclusions are entirely anatomical: to determine whether there are anatomical differences between injection of AAV2retro into the gray matter and injection into the dorsal white matter, it is important to confirm that all analyzed datapoints indeed represent injection of AAV2retro into the target region.

Comment 6: “Cell counts are derived from a single medial and lateral slice per hemisphere, which may not capture regional variability; normalising counts to cortical volume or applying stereological sampling would provide a more reliable quantitative picture.”

Response 6: Because we aim to specifically target gene delivery to sensorimotor corticospinal neurons, ideally the neuron counts in Figure 2 would be normalized by the total number of corticospinal neurons in each section. Although we agree that there may be variability in the total number of corticospinal neurons in each analyzed brain section, and that quantifying the percentage of corticospinal neurons that are transduced in each section would improve reliability, we unfortunately are not aware of any methods that can accurately quantify the number of corticospinal neurons in each section. We searched the literature for corticospinal neuron-specific immunolabeling markers, but we did not find any markers that are specific for corticospinal neurons in mouse sensorimotor cortex. Because the number of corticospinal neurons in each section may not scale proportionally with cortical volume, we do not believe it would not be appropriate to normalize these counts by cortical volume.

We attempted to partially address this concern by analyzing both medial (Figure 2A) and lateral (Figure 2B) sensorimotor cortex. We observed nearly identical effect sizes in both regions despite differences in the raw number of quantified neurons. Compared to injection of AAV2retro into intact gray matter, injection of intact white matter increased the number of tdTomato-labeled neurons by an average of 215.2 per medial brain section  (d = 5.15 [161.4, 269.0] and by 185.2 per lateral section (d = 5.16 [139.0, 231.3]). The observed differences between groups thus appear to be highly consistent across different levels of sensorimotor cortex, despite variability in the total number of corticospinal neurons at each level.

We also performed a new analysis of tdTomato-labeled corticospinal axons in L4 lumbar spinal cord, shown in Figure 3 and described in lines 392-404  After injection of AAV2retro into the intact C4 gray matter, very few tdTomato-labeled axons were observed in the lumbar dorsal columns (Figure 3A). In contrast, after injection of AAV2retro into the intact C4 dorsal column white matter, significantly more tdTomato-labeled hindlimb corticospinal axons were observed within the lumbar dorsal columns and innervating the lumbar gray matter (Figure 3B). No labeled corticospinal axons were observed in the lumbar spinal cord in mice that received a C5 dorsal column lesion (Figure 3C,D), consistent with complete transection of the corticospinal tract.  Unlike the brain analysis shown in Figure 2, quantifying the L4 dorsal column white matter samples the entire population of corticospinal neurons that project below L4, reducing the potential for regional variability among tissue sections.

In summary, we acknowledge that our methods cannot quantify the exact percentage of corticospinal neurons that are transduced in each group, because we are not aware of any method that can provide the number of corticospinal neurons in each brain section. Nonetheless, our results are consistent across multiple levels of sensorimotor cortex and lumbar spinal cord, and strongly support our primary conclusions that injection of AAV2retro into the intact dorsal column white matter targets both forelimb and hindlimb corticospinal neurons and substantially increases the number of transduced corticospinal neurons relative to gray matter injection.

Comment 7: “The study focuses on histological transduction without behavioural or electrophysiological measures, limiting insight into functional relevance and translational potential.”

Response 7: Future studies will examine therapeutic efficacy of regenerative gene delivery in SCI models by injection of AAV2retro into the dorsal column white matter, with outcome measures including corticospinal axon regeneration and functional recovery. However, several recent studies have suggested that regenerative gene therapies must be transiently expressed in corticospinal neurons only until axon regeneration has restored the descending motor circuitry, and that prolonged upregulation of pro-growth signaling can cause toxicity and adverse side effects (Campion et al. 2022, PMID: 34953897; Metcalfe et al. 2023, PMID: 37778650; Stewart et al. 2023, PMID: 37558155). We are actively researching translational methods for temporal control of regenerative gene expression, such as the “Xon” system, which uses drug-induced alternative splicing of AAV mRNA to control protein translation (Monteys et al. 2021, PMID: 34321659). We intend to deliver such drug-inducible therapies by AAV2retro injection into the dorsal column white matter, which could potentially improve safety and efficacy by specifically targeting gene delivery to corticospinal neurons while also controlling the strength and duration of regenerative gene expression. This topic is discussed in lines 605-624.

Comment 8: “All mice are four-month-old females and the manuscript does not describe randomisation or blinding during surgery or analysis, leaving room for unconscious bias and constraining external validity.”

Response 8: Blinding of the surgeon to group identity during surgery was not possible, because each group received a unique surgical procedure. Animals were randomly assigned to each group at the time of surgery. We added this information to the Methods, line 137.

Blinding prior to analysis is described in the Methods, lines 221-224 and 269-270.

Comment 9: “No raw count spreadsheets, analysis code, or detailed image-processing workflow is shared, so independent verification and replication are not currently possible; depositing the ImageJ macros and statistical scripts would markedly improve transparency.”

Response 9: All raw data are included in the manuscript. For the graphs shown in Figure 2, Figure 3, Supplemental Figure S2, and Supplemental Figure S3, each dot represents an individual datapoint, and every datapoint is shown. We added this statement to the Methods, lines 224-225.

We did not use any custom code or statistical scripts for these analyses. All analyses were performed in ImageJ or Zeiss Zen 3.1 (blue edition) software using built-in software commands. To support independent verification and replication, we added a detailed description of each analysis method to lines 231-234, lines 246-259, and lines 271-280.

Reviewer 3 Report

Comments and Suggestions for Authors

Interesting work, well presented. Interesting investigation of rostral injections following complete SCI in mice, but only immediately after injury. Compares injection into gray and white matter, and follows retrograde 'targeting' and expression of markers/ line 56, is direct injection of AAV into brain 'safe'? line 74, what about non viral delivery, which has been shown to achieve robust expression. How does a rostral injection traverse the injury? Certainly after more that a few minutes to hours after injury the post injury changes have been the major impediment to delivery of any presumed therapeutic to minimize the secondary injury or to promote regrowth. N=8  Numbers of animals are small. Why independently examine each side of sc and brain? Not sure that the rationale given is convincing. line 252  I'm not sure this can really be described as 'targeting transected axons'. line 254 on  Is this more Discussion? All in all well written and well presented, but as a Neuro Anesthesiologist I have a major problem with the experiment as performed. The time course for injection of even the most robust protective and/or therapeutic injection after SCI will be be hours at best, and then ideally it would be into CSF and NOT requiring precise sc injection requiring GA, positing, stereotaxic frames and MRI or CT guidance. The time course after injury for SCI and for TBI is very multi factoral, and changes dramatically from minutes to hours to days to weeks after the initial insult. These are interesting results, but perhaps searching for a target with an appropriate time course for realistic clinical application, and requiring more planning and discussion. 

Author Response

Thank you for these helpful comments. Please see point-by-point responses below.

Comment 1: “Line 56, is direct injection of AAV into brain 'safe'?”

Response 1: Previous clinical trials have safely injected AAV vectors into the human brain (Rafii et al. 2014, PMID: 24411134; Van Laar et al. 2025, PMID: 40395017). However, as noted by the reviewer, this approach is not inherently safe and requires substantial expertise to be safely performed. We revised this sentence to more accurately state that “direct AAV injection into the human brain can be safely performed” (line 52).

Comment 2: “Line 74, what about non viral delivery, which has been shown to achieve robust expression.”

Response 2: Non-viral methods for nucleic acid delivery are appealing because they transiently alter gene expression and could potentially promote regeneration of injured neurons within a limited therapeutic window, but these approaches also have drawbacks that limit their use in SCI. Antisense oligonucleotides can transiently suppress one or more genes after infusion into the cerebrospinal fluid, but efficacy may be limited by incomplete gene knockdown (Mortberg et al. 2023, PMID: 37188501). Lipid nanoparticles carrying mRNA can drive transient gene expression, but delivery of lipid nanoparticles across the blood-brain barrier remains challenging (Khare et al. 2023, PMID: 37150326). Similar to viral gene delivery, both antisense oligonucleotides and lipid nanoparticles broadly treat the nervous system after infusion into the cerebrospinal fluid or blood. Although primary motor cortex could be treated by direct injection, this would require a large number of invasive brain injections. Thus despite recent advances in viral and non-viral gene therapies, there is currently no translational method for gene delivery to corticospinal neurons that is both broad and specific. Improved methods are needed that effectively target upper motor neurons throughout the cortex and limit off-target gene delivery to other regions.

We added the above discussion of non-viral gene therapy methods to lines 69-79.

Comment 3: “How does a rostral injection traverse the injury? Certainly after more than a few minutes to hours after injury the post injury changes have been the major impediment to delivery of any presumed therapeutic.”

Response 3: We did not predict that AAV2retro injections would traverse the injury. In the two groups of mice that received AAV2retro adjacent to SCI, the dorsal columns were transected with a wire knife, and AAV2retro was injected ~0.2 mm rostral of the transection. We hypothesized that AAV2retro injected into the dorsal column white matter at this location would spread to the rostral edge of the injury and enter the transected ends of corticospinal axons. Although this approach was effective, such precise placement and timing of AAV2retro injection does not appear to be necessary: injection into the intact dorsal column white matter without an injury also effectively targeted both forelimb and hindlimb corticospinal neurons. This suggests that following a spinal cord injury, AAV2retro could be injected into the cervical white matter distal from the injury site to avoid any adverse impact of the post-injury environment and prevent leakage of AAV into the lesion site. We added these details to the Conclusions, lines 626-634.

Injection of AAV2retro into the dorsal column white matter is a promising method for regenerative gene delivery to corticospinal neurons, but it is well established that without further intervention regenerating corticospinal axons cannot efficiently traverse sites of spinal cord injury (Tedeschi & Bradke 2017, PMID: 28039763; de Freria et al. 2021, PMID: 34943804). Therapies that promote regeneration of corticospinal neurons must be combined with additional treatment of the injury site to provide a permissive environment for corticospinal axon regeneration, such as implantation of a neural stem cell graft or bioengineered scaffold (Kadoya et al. 2016, PMID: 27019328; Koffler et al. 2019, PMID: 30643285). Although further preclinical studies are needed, it may be possible to implant a stem cell graft or scaffold into the injury site and inject AAV2retro rostral of the injury during the same surgical session. We added a discussion of this point to lines 570-577.

Comment 4:  “N=8  Numbers of animals are small. Why independently examine each side of spinal cord and brain?”

Response 4: We added more extensive justification supporting the analysis of each side of the brain or spinal cord as an independent datapoint (lines 304-311). In rodents, corticospinal neurons on one side of the brain almost entirely project to the contralateral side of the spinal cord. Each spinal injection of AAV2retro thus separately targeted a distinct population of corticospinal neurons on the opposite side of the brain. Because we exclusively examined anatomical outcomes in this study, we analyzed each anatomically independent side of the brain or spinal cord as an independent datapoint. The results supported the validity of this approach: substantial differences between the left and right sides were observed in some mice, indicating that the two sides are anatomically distinct circuits that may be differently targeted by the separate AAV2retro injections on each side. We added Supplemental Figure S2 to directly highlight differences between the left and right sides in a mouse treated by injection of AAV2retro into the intact dorsal column white matter. In this mouse, substantial differences between the left and right sides were observed both in the number of tdTomato-labeled corticospinal neurons in sensorimotor cortex and in the number of tdTomato-labeled corticospinal axons in the lumbar dorsal columns. We believe this strongly supports the analysis of each side as an anatomically independent circuit. Supplemental Figure S2 is now described in lines 405-411 and included in the supplementary material.

Comment 5:  “Line 252:  I'm not sure this can really be described as 'targeting transected axons'.

Response 5: Although we initially hypothesized that injection of AAV2retro rostral of SCI would increase retrograde transduction of corticospinal neurons by targeting transected axons, we agree that our results do not directly support this hypothesis. To avoid potential confusion, we removed this reference to “transected axons” from line 356 (previously line 252).

Comment 6:  “Line 254: Is this more Discussion?”

Response 6: We believe it is important to briefly address the observed variability in transduction of corticospinal neurons after gray matter injection of AAV2retro adjacent to SCI (Figure 2). This could potentially reflect variable access to transected corticospinal neurons due to variable spread of AAV2retro from the gray matter injection site into the spinal cord lesion. We address this point in the Results because it directly informs the interpretation of Figure 4, where we describe variable tdTomato immunolabeling in spinal cord sections after gray matter injection adjacent to SCI, which again is potentially due to variable spread of AAV2retro into the spinal cord lesion (see Figure 4C and lines 429-439).

To improve clarity and more explicitly link these two points, we added subheadings to the Results section, and we added a note at the end of line 363 (previously line 254) to “see Section 3.4, below.”

Comment 7:  “The time course for injection of even the most robust protective and/or therapeutic injection after SCI will be be hours at best, and then ideally it would be into CSF and NOT requiring precise sc injection requiring GA, positing, stereotaxic frames and MRI or CT guidance. The time course after injury for SCI and for TBI is very multi factorial, and changes dramatically from minutes to hours to days to weeks after the initial insult. These are interesting results, but perhaps searching for a target with an appropriate time course for realistic clinical application, and requiring more planning and discussion.”

Response 7: See also response 3 (above). Precise placement and timing of AAV2retro injection relative to the injury site does not appear to be necessary: injection into the intact dorsal column white matter without an injury also effectively targeted both forelimb and hindlimb corticospinal neurons. This suggests that following a spinal cord injury, AAV2retro could be injected into the cervical white matter distal from the injury site to avoid any adverse impact of the post-injury environment and prevent leakage of AAV into the lesion site. After SCI, transected corticospinal axons typically retract from the injury site and form terminal swellings known as retraction bulbs (Hill 2017, PMID: 27825985). In mice, deletion of the PTEN gene one year after SCI can effectively promote corticospinal axon regeneration, indicating that chronically injured corticospinal neurons can still respond to regenerative gene therapies (Du et al. 2015, PMID: 26134657). Although further research is needed, these findings suggest that regenerative gene delivery by injection of AAV2retro into the spinal white matter could potentially promote corticospinal axon regeneration even when injected distal from the lesion site in chronically injured subjects.

We added these details to the Conclusions, lines 626-647. As noted in the Discussion, although extensive preclinical research will be needed to determine whether this new approach for gene delivery to corticospinal neurons is clinically feasible, we believe the promising results of this mouse study strongly support further research in preclinical models of SCI.

Reviewer 4 Report

Comments and Suggestions for Authors

Only one minor comment. This is a very good paper. Figure 1 is clearly depicted the experimental design. Aa each side of the brain is separately targeted by an independent AAV2retro injection on the corresponding side of the cord, are the mark Xs in the figure indicated at the AAV2retro injection sites? You may double check the X locations indicating white matter injection.

Author Response

Comment 1: “Figure 1 is clearly depicted the experimental design. Are the mark Xs in the figure indicated at the AAV2retro injection sites? You may double check the X locations indicating white matter injection.”

Response 1: Thank you for this comment. Yes, in Figure 1 the sites of AAV2retro injection are indicated by an “X” symbol. We added this statement to the Figure 1 legend (line 316).

We also modified the schematic images of spinal cord in Figure 1 to better represent the anatomy of mouse cervical spinal cord,  and we have confirmed that the injection sites are now accurately indicated on these updated schematics.

Round 2

Reviewer 2 Report

Comments and Suggestions for Authors

Authors have addressed suggestions from previous round of review.

Reviewer 3 Report

Comments and Suggestions for Authors

adequate responses to my first review